# The Link between Food Environment and Colorectal Cancer: A Systematic Review

**DOI:** 10.3390/nu14193954

**Published:** 2022-09-23

**Authors:** Noor Azreen Masdor, Azmawati Mohammed Nawi, Rozita Hod, Zhiqin Wong, Suzana Makpol, Siok-Fong Chin

**Affiliations:** 1Department of Community Health, Faculty of Medicine, Universiti Kebangsaan Malaysia, Kuala Lumpur 56000, Malaysia; 2Department of Medicine, Faculty of Medicine, Universiti Kebangsaan Malaysia, Kuala Lumpur 56000, Malaysia; 3Department of Biochemistry, Faculty of Medicine, Universiti Kebangsaan Malaysia, Kuala Lumpur 56000, Malaysia; 4UKM Medical Molecular Biology Institute, Universiti Kebangsaan Malaysia, Kuala Lumpur 56000, Malaysia

**Keywords:** colorectal cancer, food availability, food environment, incidence, mortality

## Abstract

Food and diet are critical risk factors for colorectal cancer (CRC). Food environments (FEs) can contribute to disease risk, including CRC. This review investigated the link between FEs and CRC incidence and mortality risk. The systematic search of studies utilised three primary journal databases: PubMed, Scopus, and Web of Science. Retrieved citations were screened and the data were extracted from articles related to the FE-exposed populations who were at risk for CRC and death. We evaluated ecological studies and cohort studies with quality assessment and the Newcastle-Ottawa Quality Assessment Form for Cohort Studies, respectively. A descriptive synthesis of the included studies was performed. Out of 89 articles identified, eight were eligible for the final review. The included studies comprised six ecological studies and two cohort studies published from 2013 to 2021. Six articles were from the US, one was from Africa, and one was from Switzerland. All eight studies were of good quality. The significant finding was that CRC incidence was associated with the availability of specific foods such as red meat, meat, animal fats, energy from animal sources, and an unhealthy FE. Increased CRC mortality was linked with the availability of animal fat, red meat, alcoholic beverages, and calorie food availability, residence in food deserts, and lower FE index. There were a variety of associations between CRC and the FE. The availability of specific foods, unhealthy FE, and food desserts impact CRC incidence and mortality. Creating a healthy FE in the future will require focus and thorough planning.

## 1. Introduction

Colorectal cancer (CRC) is the third most common cancer and accounts for more than 1.9 million new cases after breast and lung cancer [1]. According to GLOBOCAN 2018, CRC is the third most prevalent cancer in men and the second most common cancer in women. The age-standardised CRC incidence rate was 19.5%, which was higher than that of other cancers. The rapid rise in CRC incidence was associated with urbanisation and lifestyle westernisation, which caused a change in eating behaviour [2] that increased obesity prevalence [3,4,5]. Furthermore, dietary factors significantly contributed to CRC mortality risk [6]. Nevertheless, CRC is highly preventable [7]; therefore, an important CRC prevention measure is to improve the modifiable CRC risk factors in diet and nutrition.

The Committee on World Food Security stated that the food environment (FE) consists of the physical, economic, political, and sociocultural contexts in which consumers interact with the food system to attain, prepare, and consume food [8]. Diet patterns are strongly related to the FE based on availability, affordability, cost, and sustainability [2]. The FE and diet patterns were linked to increased risk of non-communicable diseases such as obesity, coronary heart disease, and diabetes [9]. The FE accessibility can be influenced by the population’s physical, social, cultural, economic, and policy conditions and can impact the population’s food selection, pattern, and quality. Many studies examined specific food items, such as red meat or fibre, and CRC incidence in diverse settings [10,11,12,13,14]. There must be interplay by the FE to affect CRC risks and outcomes [15]. Although numerous researchers used a population’s FE to evaluate potential correlations with disease [16], there remains a lack of studies investigating the link between FEs and CRC. This review will discuss recent findings and answer the following questions: What is the FE, and does the population exposed to or living in the FE risk CRC and death? This paper fills the gaps of prior reviews and adds to knowledge on the FE and CRC, which may help developed and developing countries improve their FEs and minimise CRC risk.

## 2. Materials and Methods

This review is registered under PROSPERO (CRD42022326513).

### 2.1. Research Question Formulation

The review was guided by population, exposure, and outcome (PEO) to determine the relevant articles [17]. The keywords were created based on the research questions and searched using related references, online thesauruses, and dictionaries. The keywords used are shown in Table 1.

The search strategy followed the Preferred Reporting Items for Systematic reviews and Meta-Analyses (PRISMA) checklist. Figure 1 depicts the systematic article review process.

### 2.2. Identification

We searched for articles focusing on the FE and the related CRC outcomes in PubMed, Scopus, and Web of Science. A preliminary search was conducted to identify the appropriate keywords and determine whether the review was feasible, and identified 89 titles.

### 2.3. Screening

After removing 17 duplicate articles, we screened 72 titles and abstracts based on the inclusion and exclusion criteria. The inclusion criteria were: full-text original article obtained via open access or institutional subscription. Both observation and intervention study designs that answered the review questions were accepted. The following article types were excluded: (a) case studies, (b) systematic and narrative review papers, and (c) non-English articles. Non-English articles were excluded from this review as the use of such articles might have involved increased costs, time, and skills. Sixty-two articles were eliminated after the title and abstract screening, leaving 10 articles for evaluation of eligibility.

### 2.4. Eligibility

The full text of the 10 remaining articles was obtained and thoroughly reviewed to determine whether the article met the inclusion criteria and objectives. Each author compared their list of potentially relevant articles and discussed their selections until an agreement was reached. Two articles that did not answer the research question were excluded. Therefore, a total of eight articles were accepted.

### 2.5. Data Extraction

The data from the eight included articles were entered into Excel (Microsoft Corporation, Redmond, WA, USA) and customised according to (a) year of publication (b) authors, (c) country, (d) title, (e) study design, (f) FE, (g) CRC outcome, and (h) conclusions.

### 2.6. Data Analysis

The data were primarily described using a narrative approach. The results relevant to the review questions were extracted and organised in a data extraction form with informal methods by ordering the tables and figures used to investigate heterogeneity, and concluded to prioritise result summarisation. The findings included the number of studies and participants, risk of bias in the studies, study directness in addressing the review questions, and the risk of publication bias to address the certainty of the evidence. The data were presented in tables to compare the findings of the included studies. Two reviewers double-checked the accuracy of the extracted data independently. To ensure that the results were consistent, all reviewers were required to agree on a result reporting structure.

### 2.7. Quality Appraisal

The design quality of the included ecological studies was analysed using modified criteria adapted from Dufault and Klar [18] and Betran [19] as there is no validated tool for such studies. Each ecological study scored a maximum of 21 points for 15 elements: 12, 6, and 3 points for study design, statistical methods, and reporting quality, respectively. All included studies were awarded a star for the questions answered in the respective domains of the checklist. The included cohort studies were evaluated using the Newcastle-Ottawa Quality Assessment Form for Cohort Studies. All included cohort studies were awarded a star for the questions answered in the respective domains of the checklist. An article was included in the review if both reviewers agreed on its quality. The assigned reviewers addressed any disagreement by consulting a third independent reviewer. The eight included studies had scores of 16–18 out of 21, which indicated that they were of adequate quality. The quality assessment is depicted in Table 2 (cohort studies) and Table 3 (ecological studies).

## 3. Results

The initial search returned 89 studies. The duplicates were removed, and the titles and abstracts of the remaining articles were screened to determine if they answered the research questions. Due to the considerable heterogeneity, the data are presented descriptively.

### 3.1. Characteristics of the Included Studies

The eight articles included in this review were published from 2013 until 2021. Six articles were ecological studies and two were cohort studies. Six articles were from the USA, one was from sub–Saharan African countries, and one was from Switzerland. Table 4 presents the characteristics of the included studies.

### 3.2. FE Attributes

The FE attributes were availability of unhealthy food, restaurant environment index (REI), trend changes in fast foods and supermarkets, food deserts, availability of specific foods, and the FE index (FEI). Table 5 presents the description and interpretation of the FEs.

The FE outcomes were categorised as the CRC incidence and mortality rate. Five studies reported on the FE effect and CRC incidence while two reported on CRC mortality. The findings are summarised in Table 6.

#### 3.2.1. FE and CRC Incidence

Two studies reported a significant link between CRC incidence and the FE. Mo et al. reported a significant link between a low FEI and CRC incidence while Aglago et al. [26] reported that CRC in men and women was significantly positively correlated with red meat, meat, animal fats, and availability of energy from animal sources. Conversely, Gibson et al. [22] reported that CRC incidence was not strongly associated with increased availability of unhealthy foods. There was also no association between CRC risk and obesogenic neighbourhood attributes such as the REI and retail FEI (RFEI) [20]. Besson et al. [24] found no association between FE and CRC incidence, but further investigation revealed associations between polyp growth and fish availability and decreased availability of animal fats.

#### 3.2.2. FE and CRC Mortality

The CRC mortality rate and availability of animal fat, red meat, alcoholic beverages, and calories were strongly related [25]. Higher CRC mortality was significantly associated with living in a food desert [23] and a lower FEI [15].

## 4. Discussion

In this study, eight articles on CRC and the FE were reviewed. Four studies [15,23,25,26] reported on the relationship between the FE and CRC. Broadly, FEs were divided into natural FEs (wild or cultivated) and built FEs (informal and formal markets) [27]. The FEs varied according to culture, tradition, habits, race/ethnicity, and setting (urban or rural) [28]. Varied results were attributed to the study population, FE attributes used, and variations in the FE and CRC measurements.

Mo et al. [15] described the association between poor FEs and CRC incidence. Poor (lower) FEI localities were characterised by limited access to healthy foods, lower annual income, farther distance from grocery shops, and unreliable food source. The CRC mechanism with underlying poor FE remains unknown. The availability of fast food outlets was associated with high body mass index (BMI), body fat, obesity, and frequent processed meat consumption, which were closely related to CRC risk factors [29]. Fewer studies investigated the relationship between built FEs (fast food restaurants and grocery shops) and CRC. An ecological study investigating diabetes prevalence reported that decreased diabetes prevalence was associated with grocery shops and full-service restaurants (seated and pay after eating) while high diabetes prevalence was associated with fast food restaurants [30]. Among Mexican adults, accessibility to grocery shops in food-insecure communities might have increased the likelihood of obesity [31] and led to a higher mean BMI [32]. Poor FEs affected the incidence of CRC and other non-communicable diseases, which emphasised the importance of recognising poor FEs and necessitating multidisciplinary perspectives and approaches.

Food deserts are characterised by low access to healthy food and the presence of low-income areas [33]. Food deserts and food accessibility are notably influenced by distance, race/ethnicity, income, and age [34]. Inaccessibility referred to barriers in the locality, such as accessibility to healthy food and personal barriers such as financial barriers, lack of transportation (public or personal), or below-average family income. Fong et al. [23] reported the association between food deserts and CRC mortality. Food deserts were also linked with heart disease [33] and increased risk of all-cause and cardiovascular hospitalisation [35]. Due to a lack of access to food markets that sold reasonably priced nutritious foods, poor communities were more likely to consume processed foods, refined grains, and fewer fresh vegetables [15]. Apart from food, poor communities also encountered difficulty accessing health facilities such as hospitals, clinics, and pharmacies, which could affect their health and disease outcomes [23]. Although socioeconomic inequalities were weakly associated with non-communicable diseases and risk factors [36], equal access to healthy foods in impoverished neighbourhoods must be highlighted [37]. An established healthy FE would support healthy eating and improve population health [38].

The availability of specific foods, such as red meat, animal fat, and energy from animal sources, was associated with CRC incidence [26] and mortality [25]. The findings were consistent with that of Hoang et al. [6], who reported a positive link between red meat and all-cause mortality among patients with CRC. Many studies reported a connection between CRC and red meat diet patterns [12,39,40,41] where fast food and westernised diets contained unhealthy combinations of red meat, processed meat, sugary drinks and desserts, and processed snacks, which have all been linked to gut inflammation [42]. The formation of mutagenic and carcinogenic agents in red meat was linked to the disruptions in homeostasis and colonic epithelial cell renewal that lead to CRC [43]. Increased availability and consumption of animal-derived products and a concomitant reduction in the traditional plant-based diet may drive the rising incidence of CRC in many sub-Saharan African countries. Most African countries are transitioning rapidly from traditional foods to animal-sourced foods and highly processed diets, increasing diet-related non-communicable diseases and cancers [26].

Buamden [25] reported that high alcohol drink availability was associated with CRC mortality. Alcohol consumption may be influenced by its widespread availability. Many studies reported the association of alcohol with CRC [4,44,45,46], specifically, people who consumed at least four daily drinks were more likely to develop CRC than non-drinkers [47]. Alcohol intake may initiate carcinogenic processes by destroying folate when microbially converted into acetaldehyde in the colon. Subsequently, the folate deficiency results in chromosome breakage, uracil misappropriation, and other DNA precursor abnormalities, which initiates CRC [48]. Empowering consumers by providing health education and promoting healthy food choices may help reduce the impact of the high availability of unhealthy food. Moreover, a local framework can be proposed to facilitate FE monitoring [49].

FE may be linked to ethnicity, socioeconomic and environmental factors, resulting in CRC risk [15]. Mo et al. discovered stronger CRC associations in areas with a higher proportion of Black populations. According to Carethers [50], factors contributing to ethnic and racial disparities in CRC include genetic and environmental susceptibility (a high red meat, fat, or calorie diet, obesity, a low-fibre diet), and low screening utilisation. Previous research has limited evidence of the link between CRC, screening behaviour, and FE. Screening reduces incidence and mortality by 50% and 53%, respectively, whereas primary prevention can fill the remaining [51]. Because of its ability to completely visualise the colon, colonoscopy is diagnostic and therapeutic. The screening target in the United States is 80%; however, disparities in screening utilisation across US subpopulations may contribute to CRC disparities [52]. Black and Hispanic Americans have the lowest screening rates with a family history. Black Americans are less likely to be aware of their parent’s cancer history than White Americans, and family members are less likely to report colonic polyps. Lack of provider recommendation for screening, fear of diagnosis, scheduling, implementation of screening, inability to pay for the colonoscopy due to economic difficulties, and loss of follow-up are likely barriers to colonoscopy utilisation [52]. CRC incidence and mortality could be effectively reduced through primary and secondary prevention. In addition to screening, adopted healthy behaviours (alcohol consumption, smoking, physical activity, BMI, and diet) [53] are associated with a lower risk of colorectal adenoma and higher adherence to a healthy lifestyle [54] associated with a lower risk of CRC. Previous research among Asians in California observed the strong relationships between CRC incidence, nativity and ethnic community, suggest a prominent role of acquired environmental variables such as FE [55]. The consequences of lifelong biological differences, as well as the effects of missed screening in populations, raise the risk of CRC [50].

FE may influence the pathogenesis and development of CRC. Cancer is not a single entity, and its causes are multifactorial. Significant advances in molecular carcinogenesis found diverse mutagenic events ranging from single-base substitutions to more extensive structural genetic alterations. Non-mutagenic environmental exposures interact with cellular processes and affect endogenous tumour mutations. Life-course events in the macro and micro-environments may leave genetic or epigenetic modifications expressed later [56].

Chronic inflammation has been linked to cancer. An unhealthy diet and alcohol consumption can contribute to chronic systemic inflammation and cancer cell growth. Early onset CRC is frequently associated with inflammatory bowel diseases such as Crohn’s disease and ulcerative. Molecular studies also revealed that a poor diet produced pro-inflammatory cytokines and a slew of free radicals at the cellular level, with potential gene-environment interactions in the colorectum [57,58]. Foods with a high dietary inflammatory index score were linked to an increased risk of CRC [59]. 

There were shreds of evidence implicating a robust effect of a diet that may put young, non-obese and healthy people at risk of CRC [42,60]. Another CRC risk factor is age, with CRCs increasing after age 50, but recent trends indicate more early-onset CRC [61]. In the absence of traditional hereditary factors, genetic abnormalities conferring increased susceptibility and environmental factors are likely to play a role in young-onset CRC [62]. Fast food consumption with obesity, type 2 diabetes, metabolic syndrome, and the smoking trend has increased among young people, reduces the age at which CRC develops [62]. The gut microbiota may likely occur at the intersection of these risk factors and young CRC [58,63]. Breastfeeding, diet, and obesity affect microbiome composition, increasing CRC risk in younger adults [64]. The westernisation of diets characteristically includes a high intake of red and processed meats, high-fructose corn syrup, unhealthy cooking methods, stress, antibiotics, synthetic food dyes, and monosodium glutamate are key risk factors [60]. Studies suggested that carcinogenic chemicals such as heterocyclic amines (HCAs), polycyclic aromatic hydrocarbons (PAHs), and acrylamides produced during food preparation nowadays increase CRC risk [65,66,67]. Examining foods and their environment that account for the interaction of several nutrients may shed light on the role of diet in colorectal carcinogenesis among young people. 

According to a systematic review by Puzzono et al. [60], the interaction of genetic and environmental factors is still unknown. A study supported Japanese high CRC risk relative to Caucasians by genetic susceptibilities [68]. However, studies have proposed that diet partially explains the degree of variation; for example, the risk of CRC is not distributed evenly among the four Lynch syndrome genes. Thus, diet may explain an increased or decreased risk of CRC [60]. The risk of CRC is modifiable and predominantly environmental according to temporal trends and migrant studies [69]. A paper by Maskarinec and Noh [70] analysed the cancer incidence trends among Japanese in Japan and Japanese and Caucasians in Hawaii between 1960 and 1997, highlighting that migration’s impact on cancer risk was substantial for CRC. The wide range of time migrants adopts the host population’s cancer risk demonstrates that risk factors have organ-specific effects or work at different times in life. The disparity in cancer incidence across generations suggests that staying in the host country is insufficient to increase cancer risk to the host population. Although known etiologic factors can partly explain migration, much of the variable risk remains unknown. A study found that comparing Okinawan ancestors in Hawaii to Japanese migrants from all other provinces increased exposure to cancer risks unique to a specific environment and discovered that Okinawan food environment risk factors, such as flour usage from cycad nuts, caused bowel tumours [71].

Literature on the link between FE and CRC is less consistent, as several studies reported no connections. Besson’s research found no association between the availability of food and the occurrence of colon cancer. Still, the availability of sugar, sweetened animal products, milk, meat, and fat are positively related to colorectal polyp incidence (precursors to CRC) [24]. Shvetsov et al. found a strong association between the change in the local obesogenic environment to the CRC risk. The high-density communities had low socioeconomic and resource inequality; thus, changes in behaviour or obesity-related stress may increase CRC risk. However, the FE attributes have no significant association with CRC risk. Significant relationships were only observed in one race/ethnicity, highlighting the need to study the impacts of neighbourhood change by race/ethnicity [20]. Gibson et al. [22] proposed possible explanations for the lack of an association between the food environment and CRC incidence: the availability of unhealthy foods may be influenced by factors such as transportation and dietary preferences. Another possible explanation for mixed findings is that the direction and strength of the association between diet and cancer incidence vary according to lag time [29]. Individual-level behavioural factors such as obesity and physical activity may play a role in the link between the environment and cancer risk [21]. Besson stated that the diverging trends for cancer incidence and mortality were due to improvements in screening diagnostic capacity and treatment [24]. More accurate approaches by measuring CRC incidence instead of mortality rate are suggested by Buamden [25]. Compared to the incidence rate, the mortality rate can be influenced by timely diagnosis and treatment, which varies by count and is not entirely influenced by diet or FE. Furthermore, it is concerned that there is a disparity in data quality reported across African countries, which may not reflect true heterogeneity [26]. The small sample size, lack of data on possible confounding factors such as sedentary behaviour, and lifestyle and environmental factors are other drawbacks [25].

There is a need for a reliable and validated standard to assess the FE in a specific area. The literature contains many FE indicators from interviews, questionnaires, checklists, and inventories [28]. Nevertheless, the usage of these indicators is not standardised, possibly because FE categorisation covers a broad area, such as food shop environments (grocery shops, supermarkets, farmers’ markets), restaurants (fast food and full-service), schools (cafeterias, vending machines, snack shops), and the workplace [28]. The FEI is a measure from the County Health Rankings produced from the University of Wisconsin Population Health Institute and is determined by socioeconomic conditions and considers the proximity to healthy foods and income. The elements include the distance an individual lives from a grocery shop or supermarket, the locations for healthy food purchases in most communities, and the inability to access healthy food due to cost barriers. Many studies utilised the FEI to determine the association between FEs and chronic diseases such as obesity and hepatocellular carcinoma [72]. In the US, the FEI aided investigations of increased cardiovascular disease (CVD) mortality associated with FE [73]. The US Department of Agriculture Economic Research Service initiated a food research atlas to assist and guide stakeholders in illustrating the impact of the FE [74] and developing standardised FE assessments according to region.

It is currently unknown as to whether changing FE will affect cancer outcomes. For example, providing access or establishing a new supermarket in a low-income neighbourhood may benefit residents’ economic well-being and health in terms of less diagnosed dyslipidaemia, arthritis and diabetes; however, it does not impact much on dietary habits [75]. Therefore, more research is needed to determine whether changing FE affects cancer outcomes.

Significant findings could help policymakers and program managers gain knowledge and lay the groundwork for future city planning to integrate a healthy FE. Policymakers could consider instituting an FE-linked policy for improving the community diet. The policy must address other FE-related issues such as nutritional composition, labelling, promotion, pricing, provision, retail, and investment [76]. In New Zealand, FE policy implementation led to excellent recommendations, such as the implementation of a national action plan for preventing non-communicable diseases, establishing priorities for reducing childhood and adolescent obesity, doubling funding for population nutrition promotion, and reducing the marketing of unhealthy foods [76]. In India, various nutrition interventions were formulated to enhance the FE through systematic planning and embracing differences in the problem for decoding and dividing to simplify resolutions to be addressed by policymakers and nutritionists in the future [77].

A nutritious diet supported by a healthy FE potentially protects against CRC development [78]. A previous review examined the benefit of a specific diet for reducing CRC incidence via food intake with a high omega 3 polyunsaturated fatty acid (PUFA) to omega 6 PUFA ratio and rich in fibre; vitamins B6, C, D, E; folic acid; selenium; and magnesium [79] and the Mediterranean diet, which is high in antioxidant properties. Nevertheless, obtaining and sustaining meals for such diet plans may be difficult [80] and largely depend on the FE. Therefore, the FE is critical in the food system as consumers can decide based on the best available options to support sustainable diets and enhance successful nutrition intervention [27].

In this review, many ecological analyses aided the determination of various FE-related risk factors that correlated with cancer incidence or mortality at the population level. Our findings may require a large epidemiological study to verify the concept. With the availability of cancer registries and improved diagnostic technologies, ecological research has become an attractive means of revealing and monitoring environmental links with cancer trends [26].

Typically, FE studies used ecological study designs to compare human health and provide a brief overview of the population and communities in developing specific disease prevention and intervention approaches. Ecological research is affordable and uses available aggregated data but may be inadequate and biased [81]. Presuming that a poor FE and diet increase the risk of CRC and mortality risk is challenging due to the different baseline nutritional status of study participants and obtaining the appropriate control groups. Therefore, the data need to be interpreted carefully.

Further cohort or case-control studies are recommended to derive definitive conclusions. Consistent methods and measurements research is also necessary to assess the association between the risk factors and outcomes. In this study, limited articles were identified due to the inclusion and exclusion criteria or the limited studies in this area. The use of only three research databases and the exclusion of non-English articles may have increased the likelihood of missing essential papers and publications.

This review focused on a modest portion of FEs rather than the broader context of physical, online, and virtual FEs (including food apps and online purchases), which may be a potential and exciting topic for future exploration.

## 5. Conclusions

This review addressed the current knowledge on the association between FEs and CRC. The findings indicated that there is a mixed relationship between CRC and FEs. The FE is essential in the food system for implementing interventions to support sustainable diets [27]. With healthy FEs, communities can choose the best option, practice healthy eating, and prevent disease. Therefore, attention and thorough planning must be used to create a future healthy FE and indirectly reduce CRC incidence and mortality.

## Figures and Tables

**Figure 1 nutrients-14-03954-f001:**
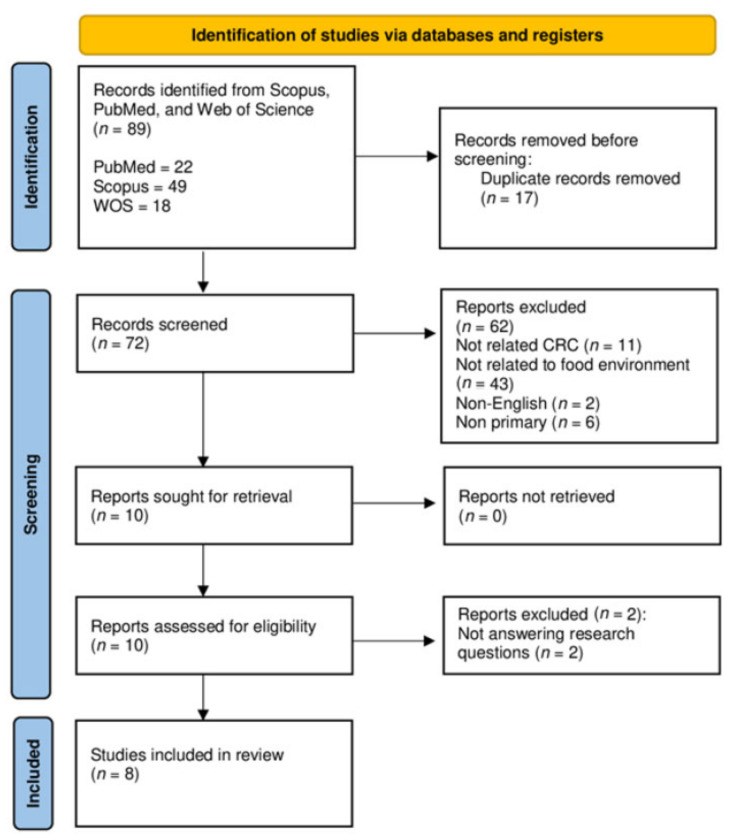
A PRISMA flow chart. WOS, Web of Science; CRC, colorectal cancer.

**Table 1 nutrients-14-03954-t001:** A keyword search used in the identification process.

Search Engine	Search Area	Search Date	Format	Search
WOS	Topic	19 April 2022	P	Colorectal cancer OR colorectal neoplasms OR colorectal carcinoma OR colorectal tumo* OR cancer colorectal OR bowel cancer OR large intestine cancer AND
			E	Food environment OR eat environment OR food dessert OR food swamp OR café OR canteen OR restaurant OR takeaway OR food entry point OR food access OR food production OR food availab* OR food access* OR food obtain OR food purchase OR food prepare* OR food handy OR food afford OR Food convenience OR food retailer AND
				Relation* OR Link OR connection OR association OR correlate OR tie AND
			O	Risk OR possibility OR probability OR frequency OR predictor OR Incidence OR occurrence rate OR frequency OR mortality OR death
SCOPUS	TITLE-ABS-KEY	19 April 2022	P	TITLE-ABS-KEY (“colorectal cancer” OR “colorectal neoplasms” OR “colorectal carcinoma” OR “colorectal tumo*” OR “cancer off colorectal” OR “bowel cancer” OR “large intestine cancer”) AND
			E	TITLE-ABS-KEY (“food environment” OR “eat environment” OR “food dessert” OR “food swamp” OR cafe OR canteen OR restaurant OR takeaway OR “food entry point” OR “food point” OR “food access*” OR “food production” OR “food availab*” OR “food access*” OR “food obtain” OR “food purchase” OR “food prepare*” OR “food handy” OR “food afford” OR “food convenience” OR “food retailer”) AND (TITLE-ABS-KEY (relation* OR link OR connection OR association OR correlate OR tie) AND
			O	TITLE-ABS-KEY (risk OR possibility OR probability OR frequency OR predictor OR incidence OR occurrence OR rate OR prevalence OR mortality or death)
Pubmed	Title/abstract	19 April 2022	P	Colorectal cancer OR colorectal neoplasms OR colorectal carcinoma OR colorectal tumo* OR cancer colorectal OR bowel cancer OR large intestine cancer AND
			E	Food environment OR eat environment OR food dessert OR food swamp OR café OR canteen OR restaurant OR takeaway OR food entry point OR food access OR food production OR food availab* OR food access* OR food obtain OR food purchase OR food prepare* OR food handy OR food afford OR Food convenience OR food retailer AND
				Relation* OR Link OR connection OR association OR correlate OR tie AND
			O	risk OR possibility OR probability OR frequency OR predictor OR incidence OR occurrence OR rate OR prevalence OR mortality OR death

WOS, Web of Science; ABS-KEY, abstract-keywords; P, population; E, exposure; O, outcome. * represents any number of characters and is used as a wildcard in keyword searches.

**Table 2 nutrients-14-03954-t002:** A quality assessment of cohort studies included in the systematic review using Newcastle-Ottawa Quality Assessment Form for Cohort Studies.

Authors	Selection				Comparability		Outcome			Total Quality Score
	Representative Eness of Exposed Cohort	Selection of Nonexposed cohort	Ascertainment of Exposure	Demonstration that Outcome of Interest Was Not Present at Start of Study	Adjust for the Most Important Risk Factors	Adjust for other Risk Factors	Assessment of Outcome	Follow-up Length Enough?	Loss to Follow-Up Rate	
Shvetsov et al. [20]	1	1	1	1	1	1	1	1	1	9
Canchola et al. [21]	1	1	1	1	1	1	1	1	1	9

**Table 3 nutrients-14-03954-t003:** A quality assessment of ecologic studies included in the systematic review.

Evaluation Criterion	Categories	Points (Max = 21)	Study					
			1 ^1^	2 ^2^	3 ^3^	4 ^4^	5 ^5^	6 ^6^
Study Design								
Design	Cross-sectional Longitudinal	1 2	1	1	1	1	1	1
Sample size	<80% units ≥80% units	0 1	1	1	1	1	1	1
Unbiased inclusion of units	No Yes	0 1	1	1	1	1	1	1
Level of data aggregation	Other than below Regional, State National	1 2 3	2	2	3	3	3	3
Level of inference	Individual or unclear Ecologic	0 1	1	1	1	1	1	1
Pre-specification of ecologic units	No Yes	0 1	1	1	1	1	1	1
Outcomes of interest included	Some All	1 2	1	1	1	2	2	2
Source of data	Inadequate Adequate	0 1	1	1	1	1	1	1
Statistical Methodology								
Analytic methodology	Spearman’s rank correlation, Linear regression model, Quadratic model, Exponential model, LOWESS, Fractional polynomial regression, Piecewise regression	1 2	1	1	1	1	1	1
Validity of regression	No Yes	0 1	1	1	1	1	1	1
Use of covariates	None Socio-economic Socio-economic + clinical	0 1	1	1	1	1	1	1
Proper adjustment for covariates	No Yes	0 1	1	1	1	1	1	1
Quality Of Reporting								
Statement of study design	No Yes	0 1	1	1	1	1	1	1
Justification of study design	No Yes	0 1	1	1	1	1	1	1
Discussion of cross-level bias and limitations	No Yes	0 1	1	1	1	1	1	1
TOTAL POINTS			17	17	17	18	18	18

^1^ Gibson et al. 2020 [22], ^2^ Fong et al. 2021 [23], ^3^ Besson et al. 2013 [24], ^4^ Mo et al. 2020 [15], ^5^ Buamden 2018 [25], ^6^ Aglago et al. 2019 [26].

**Table 4 nutrients-14-03954-t004:** Characteristics of the included studies.

Authors	Country	Study Design	Food Availability Data	Study Sample Population
Gibson et al. [22]	US	Ecological	2005 Business Patterns Survey based on matching zip codes with the US Department of Housing and Urban Development zip code	Texas Cancer Registry (5215 census tracts) Individuals aged 40, residing in Texas, diagnosed with CRC (primary/malignant and/or invasive)
Canchola et al. [21]	US	Cohort	the Restaurant Environment Index (REI) and the Retail Food Environment Index (RFEI) from California Neighbourhoods Data System	Multi-ethnic Cohort Hawaii and California
Shvetsov et al. [20]	US	Cohort	California Neighbourhoods Data System (40,870 male and 54,602 female)	Multi-ethnic Cohort Lived in California
Fong et al. [23]	US	Ecological	USDA food desert data set with zip code level measures	Stage II/III CRC patients California Cancer Registry (CCR).
Besson et al. [24]	Switzerland	Ecological	Food availability data from food balance sheets produced by the FAO	Incidence rates from the Vaud Cancer Registry.
Aglago et al. [26]	Africa (sub Saharan countries)	Ecological	Food availability data from food balance sheets produced by the FAO	African Cancer Registries Network
Buamden [25]	US	Ecological	Food availability data from food balance sheets produced by the FAO	International Agency for Research on Cancer (IARC) and codes C18, C19, C20, and C21 of the International Classification of Diseases (ICD-10).
Mo et al. [15]	US	Ecological	US FEI from the 2020 County Health Rankings	Incidence: The State Cancer Profiles by CDC and NCI 2013–2017 Mortality: CDC Underlying Cause of Mortality data 2014–2018

FAO, The Food and Agriculture Organization of the United Nations; CDC, Centres for Disease Control and Prevention; NCI, National Cancer Institute; USDA, United States Department of Agriculture; FEI, Food environment Index; CRC, colorectal cancer.

**Table 5 nutrients-14-03954-t005:** Food environment attributes in the included study, how it was measured, description and interpretation.

Authors (Year)	Attributes	Description	Interpretation
Gibson et al. [22]	Unhealthy food environment density (UFAD)	the number of all limited-service restaurants, businesses, and employment within each zip code	UFAD was divided into Quartiles1 to 4 Quartile 1 indicates the lowest unhealthy food availability Quartile 4 indicates the highest unhealthy food availability.
Canchola et al. [21]	Restaurant Environment Index	the ratio of the number of fast-food restaurants to other restaurants	-
	Retail Food Environment Index	the ratio of the number of convenience stores, liquor stores, and fast-food restaurants to supermarkets and farmers’ markets	-
Shvetsov et al. [20]	Fast food availability dynamic	Number of fast-food restaurants within blocks group	Up = increased number Down = decreased Same = similar
	Supermarket availability dynamic	Number of fast-food restaurants within blocks group	Up = increased number, Down = decreased Same = similar
Fong et al. [23]	Food desert	Areas that lack access to affordable that make up a full and healthy diet (fruits, vegetables, whole grains and low-fat milk)	Low access means: at least 500 people AND/OR at least 1/3rd of the census tract lives >1 mile in urban communities OR >10 miles in rural communities from a grocery store
Besson et al. [24]	Individual daily food availability	Estimation of the individual daily food availability of each food commodity (the total energy of animal products, vegetable products, cereals, sugars, vegetable oils, alcohol, meat, milk, fish, fruits, vegetable, fats) was made by integrating the yearly supply of domestic production+ imports + exports + stocks + non-food use, then divided by the average population and the number of days in the year to get daily availability. The values then converted to the corresponding calorie of each food commodity (kcal/person/day)	Increased total calorie of food means increased food availability
Aglago et al. [26]	Food and energy availability	Estimations of major foods and food groups available for human consumption, Total energy, proteins, fats, and carbohydrates values drawn from these food groups Data available in the food balance sheets were presented either as kilograms per capita per year or converted to kilocalories per capita per day to recover the energy contribution of the food considered.	A higher value of food and energy (in kilograms or kilocalories) means higher food availability
Buamden [25]	Food availability	Food availability represents the amount of food available per capita and provides a general picture of the populations’ diets. It does not account for food access or actual consumption	-
Mo et al. [15]	Food access	The percentage of the population that is low-income ^1^ and has low access ^2^ to a grocery store	A higher index means better food accessThe index ranges from 0 (worst) to 10 (best)
	Food security	The percentage of people without a reliable food source in the past year.	A higher index means better food security The index ranges from 0 (worst) to 10 (best)

^1^ Low income is defined as having an annual family income of less than or equal to 200 percent of the federal poverty threshold for the family size. ^2^ Access or living close to a grocery store is defined differently in rural and nonrural areas. In rural areas, it means living less than 10 miles from a grocery store whereas in nonrural areas, it means less than 1 mile.

**Table 6 nutrients-14-03954-t006:** The association between the food environment attributes and CRC.

Authors	FE	Comparator	CRC Outcome				Conclusion
			Incidence	95% CI, *p*-Value	Mortality	95% CI, *p*-Value	
Gibson et al. [22]	Quartile 2 UFA	Quartile 1	Quartile 2 1.03 ^1^	1.00–1.05	-	-	No significant differences in colorectal cancer incidence between the lowest unhealthy food availability and quartile 2,3,4
	Quartile 3 UFA	Quartile 1	Quartile 3 1.02 ^1^	1.00, 1.05	-	-	
	Quartile 4 UFA (Highest)	Quartile 1	Quartile 4 1.02 ^1^	0.99, 1.05	-	-	
Canchola et al. [21]	High REI Male Female	Low REI	0.85 ^2^ 1.29 ^2^	0.54, 1.33 0.84, 1.99	-	-	No significant associations between neighbourhood obesogenic attributes and colorectal cancer risk.
	High RFEI Male Female	Low RFEI	1.11 ^2^ 0.90 ^2^	0.91, 1.36 0.74–1.10	-	-	
Shvetsov et al. [20]	Upward change Fast food restaurants	No change	Men = 1.19 ^2^ Women = 0.96 ^2^	0.97, 1.45 (0.79–1.17	-	-	Upward change in fast food and supermarket was not statistically significantly associated with CRC risk among the male and female.
	Upward change in supermarket	No change	Men = 0.95 ^2^ Women = 1.12 ^2^	0.80, 1.13 0.96–1.30	-	-	
Fong et al. [23]	Living in Food desert Yes	No	-	-	UV HR = 1.12, MV HR: 1.18	1.05, 1.19 *p* = 0.001 1.05, 1.19 *p* = 0.001	Food desert residence was associated with higher 5-year mortality.
Besson et al. [24]	Food availability Coefficients exceeding the cut-off of ± 0.70 are considered meaningful		All types of foods results are below than 0.7	-	-	-	Colorectal cancer incidence was not associated with any food availability. Associations were found only for polyps with fish availability and decreased availability of animal fats
Aglago et al. [26]	Food availability coefficients exceeding the cut-off of ± 0.50 are correlated - T0–T20 means tumour development from 5 to over 20 years - only the significant findings showed:		Coefficient	-	-	-	Colorectal cancer incidence in men and women significantly positively correlated with red meat, meat, animal fats availability, and energy from animal sources
	Meat						
	Men		T0 = 0.72 T5 = 0.60 T20 = 0.64	-	-	-	
	Women		T5 = 0.54 T20 = 0.54	-	-	-	
	Red meat						
	Men		T20 = 0.53	-	-	-	
	Women		T0 = 0.63 T5 = 0.58 T20 = 0.58	-	-	-	
	Animal fats						
	Women		T10 = 0.67 T15 = 0.70 T20 = 0.66	-	-	-	
	Energy from animal sources		T20 = 0.52	-	-	-	
Buamden [25]	Food availability coefficients from 0.50 to 0.75 show moderate correlation and greater than 0.75 show a very good or excellent correlation.				Coefficient		Strong relationships were found between colorectal cancer mortality rate and the availability of animal fat, red meat, alcoholic beverages, and calories. The availability of fruits and vegetables have no protective effect on the colorectal cancer mortality
	Red meat		-	-	0.59	-	
	Ethanol		-	-	0.61	-	
	Total fat		-	-	0.47	-	
	Animal fat		-	-	0.60	-	
	Calorie		-	-	0.56	-	
Mo et al. 2022 [15]	Healthy FEI		41.3 per 100,000	-, *p* <0.004	14.9 per 100,000	-, *p* < 0.01	Healthy FEI scores (less food insecurity and better healthy food access were associated with lower colorectal cancer incidence and mortality A poorer food environment was significantly associated with higher colorectal cancer incidence and mortality
	Unhealthy FEI		44.5 per 100,000		17.1 per 100,000		
	Food availability coefficients		Coefficient	*p* value	Coefficient	*p* value	
	FEI		−0.681	0.004	−0.826	<0.01	
	Food insecure		−0.12	0.10	0.108	0.004	
	Limited access to healthy food		0.191	0.0001	0.096	<0.01	

UFA, Unhealthy food availability; IRR, Incidence Rate Ratio; REI Restaurant Environment Index; RFEI, Retail Food Environment Index; FEI, Food environment index; UV, Univariate; MV, Multivariate. HR, hazard ratio.^1^ Reported in incidence rate ratio (IRR). ^2^ Reported in hazard ratio (HR).

## Data Availability

Not applicable.

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
