# Peer review of "The Link between Food Environment and Colorectal Cancer: A Systematic Review"

_nutrients, 2022, doi:10.3390/nu14193954_

Round 1

Reviewer 1 Report

I am by no means an expert in Food Environments, and so please forgive my naivety in discussing your paper. The sort of studies you have ultimately selected all represent a "30,000 feet" view of diet/socio-economic status/food availablity and food choices, and their influence on the incidence and mortality of colorectal cancer. It is therefore not too surprising that 4 of the studies showed no effect of FE on colorectal cancer incidence.

In reading your discussion it seems as if you have provided something of a literature review on the topic of diet and colorectal cancer without addressing the results of your review in any great detail. Why was there no association between FE and colorctal cancer incidence in references 20,21,23,24? What does this tell us about the role of FE on colorectal cancer risk, and why did references 19,22,25 and 36 show significant associations between FE and Colorectal cancer incidence, or survival, or both? In the light of apparently conflicting data, do you propose that more research must be done? or that the positive studies were positive because of factors other than diet? or because of ethic differences? 

Just to proved a more medical background to the discussion, Bert Vogelstein, the premier scientist in the miolecular biology of colorectal cancer, has suggested that the prime driver of colorectal carcinogenesis is the high stem cell division rate in the colon. This increases the chances of pathogenic variants in driver genes which leads to loss of regulation of cell growth and division, and therefore increasing dysplasia in subclones of the affected cells. So does diet influence stem cell turnover rates? Or is the majority of colorectal cancer stochastic and therefore independant of diet? 

An alternative hypothesis may be that diet influences the immune repsonse to the dysplastic cells, and thereby permits cancers to grow that might otherwise be controlled and removed by the immune system? You have mentioned tangientially the association of diet with inflammation, another possible path to colorectal cancer. In order to make sense of the relationship between diet and colorectal cancer, if indeed there is one, it is necessary to provide a reasonable hypothesis about how this relationship would work. 

The now pretty old studies of Japanese migrants to Hawaii, and their high rates of colorectal cancer in the new environment, set that stage for these sort of discussions. (Le Marchand L, Kolonel LN. Cancer chez les migrants japonais d'Hawaï: interactions entre gènes et environnement Cancer in Japanese migrants to Hawaii: interaction between genes and environment. Rev Epidemiol Sante Publique. 1992;40(6):425-30. Stemmermann GN, Nomura AM, Chyou PH, Kato I, Kuroishi T. Cancer incidence in Hawaiian Japanese: migrants from Okinawa compared with those from other prefectures. Jpn J Cancer Res. 1991 Dec;82(12):1366-70. Le Marchand L, Wilkens LR, Hankin JH, Kolonel LN, Lyu LC. A case-control study of diet and colorectal cancer in a multiethnic population in Hawaii (United States): lipids and foods of animal origin. Cancer Causes Control. 1997 Jul;8(4):637-48.) You do not mentions these older studies and how the current research on FE relates to their findings. 

 In addition the work on Polygenic risk scores obtained by genome wide analysis studies suggests that cancer risk outside of obviously familial risk, is partly due to low incidence genetic factors. (Thomas M, Sakoda LC, Hoffmeister M, Rosenthal EA, Lee JK, van Duijnhoven FJB,et al. Genome-wide Modeling of Polygenic Risk Score in Colorectal Cancer Risk. Am J Hum Genet. 2020 Sep 3;107(3):432-444.). How does this relate to the FE studies? 

Finally it has become clear that the best way to prevent colorectal cancer is by colonoscopy. You imply that acess to colonoscopy is a factor in some of the studies you reviewed. I would say it is a critical factor. The USA is the only country where the overalll incidence of colorectal cancer is decreasing, due predominantly to the removal of adenomas facilitated by screening and surveillance colonoscopy. To a certain extent, as far as colorectal cancer is concerned, you can eat what you like as long as you get a colonoscopy. I am not sure that the obverse is true...that if you eat ony a "healthy" diet, you don't need a colonoscopy. 

The big question in colorectal circles at the moment relates to the increase in incidence of left sided colorectal cancer in people under age 50. Why is this happening? Is it dietary (possibly)? Is it hereditary (partly)? It certainly seems to represent a change in the biology of the disease and is being seen worldwide. Is there any research into the FE of people under the age of 50 as oppsed to older people? 

In summary I would like to see a more indepth discussion about the results of your review...the reasons behind the results that the reviewed articles produced, how they reconcile with each other and what they say about the role of the FE in determining colorectal cancer incidence. Finally a specualtion about how changes in the FE might influence colorectal cancer incidence. 

.

Reviewer 2 Report

This is a well-designed and a well-written article concerning the influence Food Environment on the risk of development of Colorectal Cancer, especially that availaible literature does not directly describe this issue.

 In my opinion the content of the manuscript, its aim and the direction are clear. The manuscript is original and its topic is interesting.

The title expresses clearly the content of the manuscript and highlights the importance of the study. It doesn’t contain any unnecessary description.

The abstract is a short and clear summary of the aims, key methods, important findings and conclusions of the article and doesn’t contain unnecessary information. The introduction section clearly summarize the current state of the topic as well as clearly define the aim of the study. The introduction is this consistent with the rest of the manuscript. The introduction provides the most important information about CRC and explain the definition “the food environment (FE)” .

Study design and methods are appropriate for the research question. The methods of selecting the appropriate publications has been described in detail. Keywords search used in the identification process are described in Table 1. The inclusion and exclusion criteria of papers have been described.

The results are presented clearly and accurately and are consisted with the aim of the work and the methods. All the relevant data have been included in the article. The data described in the text are consistent with the data in the figures and tables. I see two table marked as Table 1 in the text. It’s confusing.

The authors logically explain and describe their findings. The authors reviewed 8 articles assessed the relationship between the FE and CRC. The limitations of the study also have been described. The authors concluded that there is a mix relationship between CRC and FEs

The authors cite the initial discoveries where suitable. The cited studies represent current knowledge.

I suggest to accept this paper.

This is a well-designed and a well-written article concerning the influence Food Environment on the risk of development of Colorectal Cancer, especially that availaible literature does not directly describe this issue.

 In my opinion the content of the manuscript, its aim and the direction are clear. The manuscript is original and its topic is interesting.

The title expresses clearly the content of the manuscript and highlights the importance of the study. It doesn’t contain any unnecessary description.

The abstract is a short and clear summary of the aims, key methods, important findings and conclusions of the article and doesn’t contain unnecessary information. The introduction section clearly summarize the current state of the topic as well as clearly define the aim of the study. The introduction is this consistent with the rest of the manuscript. The introduction provides the most important information about CRC and explain the definition “the food environment (FE)” .

Study design and methods are appropriate for the research question. The methods of selecting the appropriate publications has been described in detail. Keywords search used in the identification process are described in Table 1. The inclusion and exclusion criteria of papers have been described.

The results are presented clearly and accurately and are consisted with the aim of the work and the methods. All the relevant data have been included in the article. The data described in the text are consistent with the data in the figures and tables. I see two table marked as Table 1 in the text. It’s confusing.

The authors logically explain and describe their findings. The authors reviewed 8 articles assessed the relationship between the FE and CRC. The limitations of the study also have been described. The authors concluded that there is a mix relationship between CRC and FEs

The authors cite the initial discoveries where suitable. The cited studies represent current knowledge.

Author Response

Thank you, please see the attachment

Round 2

Reviewer 1 Report

Thank you for responding to my comments and questions